# Effects of 1,2-Dimethylhydrazine on Barrier Properties of Rat Large Intestine and IPEC-J2 Cells

**DOI:** 10.3390/ijms221910278

**Published:** 2021-09-24

**Authors:** Viktoria Bekusova, Linda Droessler, Salah Amasheh, Alexander G. Markov

**Affiliations:** 1Department of General Physiology, Faculty of Biology, Saint Petersburg State University, Universitetskaya nab., 7–9, 199034 Saint Petersburg, Russia; a.markov@spbu.ru; 2Institute of Veterinary Physiology, Department of Veterinary Medicine, Freie Universität Berlin, 14163 Berlin, Germany; linda.droessler@fu-berlin.de (L.D.); salah.amasheh@fu-berlin.de (S.A.)

**Keywords:** colorectal cancer, 1,2-dimethylhydrazine, transepithelial electrical resistance, barrier properties, tight junction proteins, IPEC-J2

## Abstract

Colon cancer is accompanied by a decrease of epithelial barrier properties, which are determined by tight junction (TJ) proteins between adjacent epithelial cells. The aim of the current study was to analyze the expression of TJ proteins in a rat model of 1,2-dimethylhydrazine (DMH)-induced colorectal cancer, as well as the barrier properties and TJ protein expression of IPEC-J2 cell monolayers after incubation with DMH. Transepithelial electrical resistance and paracellular permeability for sodium fluorescein of IPEC-J2 were examined by an epithelial volt/ohm meter and spectrophotometry. The expression and localization of TJ proteins were analyzed by immunoblotting and immunohistochemistry. In the colonic tumors of rats with DMH-induced carcinogenesis, the expression of claudin-3 and -4 was significantly increased compared to controls. The transepithelial electrical resistance of IPEC-J2 cells increased, while paracellular permeability for sodium fluorescein decreased, accompanied by an increased expression of claudin-4. The increase of claudin-4 in rat colon after chronic DMH exposure was consistent with the acute effect of DMH on IPEC-J2 cells, which may indicate an essential role of this protein in colorectal cancer development.

## 1. Introduction

The barrier function of the large intestine is well-defined, mainly due to the selective permeability of colonocyte membranes and tight junctions (TJ) [1,2]. Pathophysiological processes in the large intestine, such as cancer, are associated with increased intestinal permeability and destruction of the intestinal barrier function [3,4,5]. Some authors have suggested that tumor formation is associated with changes in TJ permeability [6,7]. TJ reduction correlates with tumor differentiation, and TJ changes are an early and key aspect of cancer metastasis [4]. To date, several studies have shown an up-regulation of TJ proteins in colon cancer (e.g., claudin-1, -2, -3, and -4), whereas others showed no change or a down-regulation of the same claudins [7,8,9]. It is still not clear whether changes in TJ proteins are one of the reasons for neoplasm development or a consequence of carcinogenesis.

The authors have shown a change in the barrier properties of rat colons during DMH-induced carcinogenesis [10,11]. It is also still an open question whether the changes are a direct or carcinogenesis-mediated systemic effect of DMH on the barrier properties of the large intestine. A convenient model for studying the direct effect of DMH on the barrier properties of the intestinal epithelium is the non-transformed, porcine jejunal epithelial cell line IPEC-J2 [12]. The effects of DMH on barrier properties of IPEC-J2 cells have not been studied before. Comparative studies of the acute effects of DMH on TJ proteins expression in IPEC-J2 cells monolayers, and the chronic effects of DMH on the large intestine of DMH-induced rats, would make it possible to understand whether DMH directly causes a change of the barrier properties and the composition of TJ proteins of intestinal epithelial cells. We hypothesized similar changes in the expression of TJ proteins and barrier properties in both the colon of DMH-induced rats and the IPEC-J2 cells after DMH exposure.

The model of DMH-induced carcinogenesis is recognized as the most adequate model of human colon cancer, including morphological, genetic, and biochemical aspects of its pathogenesis [13,14,15]. Mapping of the colon into three types of segments—tumors, adjacent to tumors, and not adjacent to tumors—allows us to analyze the barrier properties of the entire organ in detail, making a note of the possible influence of the tumors and tumor microenvironment on the colon tissues [10,11].

The aim of this study was to investigate the expression and localization of TJ proteins in colon adenocarcinoma and non-tumor segments in the colon of rats during DMH-induced carcinogenesis and compare them with the direct effect of DMH on the barrier properties of IPEC-J2 epithelial cell monolayers.

## 2. Results

### 2.1. Western Blot Analysis of Colonic TJ Protein Expression

Expression of TJ proteins in tumor, tumor-adjacent, and non-tumor segments of the colon was determined by immunoblotting. Chronic administration of DMH revealed an increase in the expression of tightening claudin-3 (273 ± 65% vs. control, *n* = 4) and claudin-4 (222 ± 31% vs. control, *n* = 5) in tumor tissue. Other sealing claudins, namely, claudin-1 and -5, as well as pore-forming claudin-2 and occludin, did not show significant changes in all studied groups (Figure 1).

### 2.2. Localization of Tight Junction Proteins in Rat Colon Adenocarcinomas

Confocal laser-scanning immunofluorescence microscopy was performed to determine the localization of TJ proteins in the colon tissue. The signals corresponding to claudin-3 and -4 were detected both in the control and after DMH administration. In control samples, signals showed an expected localization within the colonic epithelial tissue in cell contact sites. In tumor tissue, the claudin-3 and -4 signals showed a pattern of cell disintegration. The immunofluorescent signal was mainly located in the cytoplasm of cells (Figure 2).

### 2.3. Study of TEER and Paracellular Permeability of Fluorescein in IPEC-J2 Cells

For the control group, TEER values of the IPEC-J2 cells remained stable during the whole timespan of the experiment. No significant differences were observed in controls between the first and the last day of the incubation. In the presence of 0.1 µM DMH, a gradual increase in TEER was observed, which showed significant differences in comparison with the control after 10 days (295 ± 72%, *n* = 6). Increasing concentrations revealed a dose-dependent response. 1 µM DMH induced an enhancing effect after seven days of incubation, and on the eleventh day, TEER was increased to 347 ± 45% (*n* = 6). There were no significant differences between the two experimental groups (Figure 3A).

The value of the diffusion rate of sodium fluorescein in the control was 25.7 ± 7.7 × 10^−7^ cm/s, and in the experimental groups, DMH 0.1 µM 3.7 ± 0.4 × 10^−7^ cm/s and DMH 1 µM 5.6 ± 1.2 × 10^−7^ cm/s, respectively. The paracellular permeability of IPEC-J2 monolayers after incubation with DMH in both concentrations was significantly lower than in the control (*p* < 0.05, paired *t*-test, Figure 3B).

Thus, DMH affects the monolayer of the IPEC-J2 cell line, increasing transepithelial resistance and lowering their paracellular permeability.

### 2.4. Western Blot Analysis of TJ Proteins in IPEC-J2 Cells

In IPEC-J2 cells, we investigated the expression level of TJ proteins, namely, claudin-1, -2, -3, -4, -5, and occludin, similar to the proteins studied in the colon of DMH-treated rats. Western blot analysis revealed increased levels of claudin-4 in both concentrations of DMH (138 ± 14 and 140 ± 16% vs. control, respectively; *n* = 6). At the same time, a decrease in claudin-1 was observed after incubation with 0.1 µM DMH (82 ± 7% vs. control, *n* = 6) (Figure 4). Other claudins, namely, claudin-3 and -5, as well as occludin, did not show significant changes in all studied groups. In accordance with previous studies, claudin-2 was not detectable (not shown) [16].

## 3. Discussion

Our combined in vivo/in vitro study made it possible to compare systemic and direct effects of DMH on barrier properties of the intestinal epithelium.

1,2-dimethylhydrazine (DMH) is a chemical carcinogen that causes the development of de novo tumors in the colon of rats and mice [10,17,18]. The mechanism of action of DMH is associated primarily with DNA methylation of the stem colonocytes located at the base of the intestinal crypts, with the subsequent development of colon adenocarcinomas [19,20]. DMH induces colonic neoplasms through cellular oxidative damage and upregulation of oncogenic pathways such as PI3K/Akt and Wnt [17,21]. Akt, a key player in the colon tumorigenesis, inhibits the GSK3β/APC/axin-mediated degradation of β-catenin and enhances the expression of β-catenin target oncogenes, including c-Myc and cyclin D1 [22,23].

In this study, we tested the hypothesis that tumor development under the action of DMH is accompanied by changes in the barrier properties of the colon and IPEC-J2 cells, which are based on changes in the expression of TJ proteins. Our results have provided several novel findings. (1) The barrier properties and expression of claudins are altered by DMH, both in IPEC-J2 cells and in tumors of the distal colon. (2) Claudin-4 is specifically increased by DMH compared to other claudins, which are altered differently in the colon and IPEC-J2 cells. (3) Acute and chronic effects of DMH are manifested in similar changes in the expression of claudin-4 in IPEC-J2 cells and in tumors of the distal colon, which can therefore be regarded as a common denominator in this type of pathogenesis.

According to modern concepts, neoplastic processes include suppression and destruction of intercellular contacts, loss of apical-basal polarity for the epithelium, and a loss of adhesion to the basal membrane, followed by changes in the cell phenotype or its dedifferentiation. In this regard, a priori, it would be expected that proteins involved in the formation of intercellular contacts, particularly claudins, would show a decreased expression.

However, the experimental data indicate an absence of a general down-regulation of claudin expression in neoplastic processes, as even an upregulation of some claudin family members has been reported in human cancers [24]. It is possible that claudin upregulation could come from two signaling pathways: EGFR and Wnt; both are able to increase claudin expression and are permanently activated in many cancer types, including colorectal cancer [25]. In accordance, PCR experiments revealed a high level of expression of claudin-1, -3, and -4 in tissue samples from patients with colorectal adenocarcinomas [26,27,28,29].

Until now, the effect of DMH on the barrier properties of the colon has been studied mainly by analyzing transmembrane and paracellular transport. It was shown that DMH treatment uncouples Na^+^/H^+^ and Cl^−^/HCO_3_^2−^ exchange in the distal colon in rats [30]. The authors discussed that DMH disrupts the conductance of epithelial sodium channels in the colon of rats and mice at the early stages of carcinogenesis, when no visible morphological changes have yet been observed [31,32]. Based on the literature, after treatment with DMH, carbachol depolarized the membrane voltage, probably by activating Ca^2+^-dependent Cl^−^ channels [33]. In parallel, amiloride-sensitive Na^+^ conductance was reduced [31,32].

We demonstrate that in the analysis of the barrier properties and expression of claudins under the action of DMH on the large intestine, we should emphasize other processes. The proteins that form TJs are combined into functional clusters, which are considered as a signaling platform for the regulation of cellular processes [34,35,36]. The question that still remains open for discussion is whether changes in the expression of claudins may be a trigger in the initiation of tumor transformation or a reflection of the changes in its phenotype that have already occurred in the cell.

The analysis of the expression of claudins in colon tissue and in IPEC-J2 cells showed changes in the expression of only the sealing TJ proteins: claudins-3 and -4 in the tumor and claudin-1 and -4 in IPEC-J2 cells, while the expression of other TJ proteins did not change.

Claudin-1 decreased in IPEC-J2 cells only after incubation with 0.1 μM DMH. A review of literature on the expression of claudin-1 in the development of colorectal cancer shows multidirectional changes in its expression. Most authors point out the connection between the overexpression of claudin-1 and a positive prognosis in the development of cancer [27,37], as well as the connection between its decreased level and an increase in the malignancy of neoplasms [38]. Therefore, the decrease in its level in IPEC-J2 cells under the action of DMH in our study can be interpreted as an early unfavorable change in the phenotype of these cells.

In our study, the level of claudin-3 was significantly increased in rat colon tumors. This observation contradicts the view on the putative inhibitory role of claudin-3 in the development of colorectal cancer. Genetic and pharmacological studies confirmed that claudin-3 loss induces Wnt/β-catenin activation, which is further exacerbated by Stat-3-activation and helps promote colon cancer [39]. At the same time, there is an opposite point of view. Specific inactivation of tyrosine kinase receptors, mediated by phospholipase C (PLC), as well as a signal transducer and activator of transcription 3 (STAT3), leads to a decrease in claudin-3 levels in colorectal cancer. An enhanced expression indicates an increase in the malignant potential. However, the authors note that this effect depends on the molecular subtype of colorectal cancer [40].

Claudin-4 is well-known as a component of TJ that enhances the barrier function. Thus, claudin-4 determines an increase in the barrier properties of the follicle-associated epithelium of Peyer’s patches in comparison with the adjacent villous epithelium [41]. It was found that colorectal cancer in humans exhibited significantly elevated expression levels of claudin-4 compared with normal mucosa [28]. This was associated with significant disorganization of the TJ strands and increased paracellular permeability to ruthenium red [29]. Claudin-4 appears to be elevated at the protein level in a variety of human carcinomas [24,42,43]. It has been reported that claudin-4 promoted ovarian cancer cell invasion through activating matrix metalloproteinase 2 [44]. A recent study showed that claudin-4 reinforced proliferation, invasion, and epithelial–mesenchymal transition in gastric cancer cells [45]. Moreover, silencing of claudin-4 in MCF-7 breast cancer cells resulted in a significant reduction of cell migration [46] and played a crucial role in tumor metastasis [47]. However, there are also opposite data. Loss of claudin-4 expression was correlated with poor tumor differentiation and significantly increased the likelihood of poor cancer-related survival [48].

Under the action of DMH on IPEC-J2 cells, we observed an increase in transepithelial resistance and a decrease in paracellular permeability to sodium fluorescein.

The increase in the barrier properties of IPEC-J2 cells upon their incubation with DMH was accompanied by a significant increase in the level of claudin-4, which was also significantly increased in tumors of the colon of DMH-induced rats. Thus, both in the conditions of chronic and acute exposure, the use of DMH led to a significant increase of claudin-4 in our studies.

Similar changes in the expression of claudin-4 in the colon of rats with DMH-induced carcinogenesis and the IPEC-J2 cells in our study indicate that increased claudin-4 expression may play a key role at different stages of colon carcinogenesis.

Discussing the triggering or accompanying role of claudins in the development of neoplastic growth, it should be noted that claudin-3 and claudin-4 may be receptors that trigger a cascade of changes in the epithelium, in particular for the enterotoxin Clostridium perfringens [49,50]. We should not exclude that claudin-3 and -4 may be the triggers for cell transformation. On the other hand, these claudins, as the molecular components of TJ, may be subject to changes depending on a variety of factors. Therefore, tissue exposure to cholera toxin increased the expression of claudin-3 and -4 in the rat colon [51].

Analyzing the data on the expression of claudin-3 and -4 and on the signaling pathways involved in the malignant transformation of the epithelial cells suggests that the changes in the expression of claudins may reflect triggering signaling mechanisms to contribute to cancer development.

## 4. Materials and Methods

### 4.1. Animals and DMH Treatment

Male Wistar rats aged two months and weighing 150–180 g were housed in cages (five rats in each cage) under a standard light/dark cycle (12 h light:12 h dark) at 22 ± 2 °C with ad libitum access to tap water and complete pelleted feed (Laboratorkorm, Moscow, Russia). The studies were carried out in accordance with the guidelines of the FELASA [52] and approved by the Ethics Committee for Animal Research of St. Petersburg State University (Conclusion No. 131-03-1 dated 2 February 2021).

Animals were randomly subdivided into two groups. The rats in the control group were not exposed to the carcinogen, whereas the rats in the experimental group were administered five subcutaneous injections of DMH weekly at 21 mg/kg of body weight (each dose) (Figure 5). DMH was obtained from Sigma Aldrich (Tokyo Metropolis, Japan).

### 4.2. Colon Tissues

To evaluate TJ proteins expression in the rat colon six months after the first carcinogen injection, the rats were decapitated with a guillotine (Open Science, Moscow, Russia) without anesthesia, which can cause changes in permeability [3]. Decapitation was rapidly accomplished in accordance with the recommendations of the AVMA [53].

Mapping of the colon segments in the experimental group of DMH-induced rats depended on tumor location. We took tumor, tumor-adjacent, and non-tumor segments. The tumor-adjacent segments were two segments (2 cm in length each) from both sides of the tumor. The segments which were located further are hereinafter referred to as non-tumor segments. Tumor-adjacent and non-tumor segments did not visually show morphological changes. All segments, both controls and experimental ones, were localized in the distal part of the colon.

#### 4.2.1. Western Blotting of the Colon Tissues

TJ protein expression levels were analyzed in the membrane fractions of the colonocytes by immunoblotting, as described in detail earlier [54].

Briefly, tissues were washed in PBS with calcium and magnesium and lysed in RIPA buffer, containing 25 μM HEPES pH 7.6, 25 μM NaF, 2 μM EDTA, 1% Sodium Dodecyl Sulfate (10%), H_2_O, and enzymatic protease inhibitors (completely EDTA-free, Boehringer, Mannheim, Germany), then centrifuged, and quantitative protein analysis according to Pierce (Pierce, Rockford, IL, USA) was performed by a spectrophotometer (Tecan Spectra, Tecan, Männedorf, Switzerland). Electrophoresis was performed in polyacrylamide gel using markers BenchMarkTM Prestained Protein Ladder (Invitrogen Life Technologies, Carlsbad, CA, USA). The gels were transferred to PVDF membranes with a 0.2 µm pore size (Bio-Rad Laboratories GmbH, Hercules, CA, USA), which were first incubated with primary antibodies raised against claudin-1, -2, -3, -4, -5, or occludin, and then visualized using secondary goat, anti-rabbit, and anti-mouse IgG antibodies, and chemiluminescence reaction (Bio-Rad, Hercules, CA, USA). The following antibodies were used: claudin-1 (#ab56417, 1:50, Abcam, Cambridge, UK), claudin-2 (#32-5600, 1:100, Invitrogen, Carlsbad, CA, USA), claudin-3 (#34-1700, 1:100, Invitrogen, 517 Carlsbad, CA, USA), claudin-4 (#ab53156, 1:500, Abcam, Cambridge, UK), claudin-5 (#34-1600, 1:100, 518 Invitrogen, Carlsbad, CA, USA), and occludin (#33-1500, 1:100, Invitrogen, 520 Carlsbad, CA, USA). The protein bands were detected and identified using Clarity Western ECL Substrate and the ChemiDoc Luminescence imager (Bio-Rad, Hercules, CA, USA). β-actin served as a protein-loading control, and the values were normalized on β-actin bands, respectively. The signal density in the control group was set as 100%.

#### 4.2.2. Immunohistochemistry

Embedding, sectioning, and staining of the tissue in paraffin blocks were performed as described earlier [55]. The slides were incubated with the blocking solution (PBS, 5% goat serum, 1% BSA). TJ proteins were detected by using the above-mentioned primary and secondary antibodies. The images were analyzed by using a laser scanning microscope LSM 510 META (Carl Zeiss, Oberkochen, Germany) and LSM Image Browser software 4.2 (Carl Zeiss, Oberkochen, Germany).

### 4.3. Cell Culturing and DMH Treatment

The IPEC-J2 cell culture was obtained from the DSMZ-German Collection of Microorganisms and Cell Cultures (Braunschweig, Lower Saxony, Germany). IPEC-J2 cells were cultured in 25 cm^2^ culture flasks in Dulbecco’s MEM/Ham’s F-12 (Biochrom, Berlin, Germany) containing 3.15 g/L glucose, 2 mM stable glutamine, 10% porcine serum (Sigma Aldrich, Munich, Germany), and 1% penicillin/streptomycin (Sigma Aldrich, Munich, Germany). Cells were split once a week at a ratio of 1:3. The medium was changed every 2–3 days. Cells were cultured at 37 °C in a humified 5% CO_2_ atmosphere. For electrophysiological measurements, the cells were seeded on semipermeable cell culture inserts with a diameter of 12 mm and a pore size of 0.45 μm (Millipore, Darmstadt, Germany) at a density of 3 × 10^6^ cells per multi-well plate. Routinely, 500 μL of medium was added to the apical compartment, and 1 mL of medium was added to the basolateral compartment to guarantee an equal hydrostatic pressure, as specified by the manufacturer. The incubation with DMH (Sigma Aldrich, Tokyo Metropolis, Japan) began after cells were cultured for 15 days, when the transepithelial electrical resistance (TEER) of most of them reached 2000 Ohm·cm^2^, and was carried out for 11 days. Control cells were cultured in a medium without DMH, while medium containing DMH in two micromolar concentrations (0.1 and 1 µM) was added to the basolateral compartment of experimental cells. The medium was changed every two to three days.

#### 4.3.1. Electrophysiological Assay of the Cell Culture

TEER was measured daily at the same time using an EVOM amplifier (World Precision Instruments, Sarasota, FL, USA). The TEER measurements were normalized by the plate area and resistance values of the medium. On the first day of the experimental incubation of cells with DMH, the TEER values of the samples were taken as 100%.

#### 4.3.2. Assessment of the Paracellular Permeability of the Cell Culture

Sodium fluorescein has a molecular mass of 376 Da, is electrically neutral, and diffuses through the epithelium along the paracellular path with a concentration gradient. Its permeability was measured by using the EnSpire Multimode Plate Reader (Perkin Elmer, Waltham, MA, USA) 10 days after the incubation with DMH. The initial concentration of sodium fluorescein from the apical side of the cells was 100 µM. The samples of the solutions were taken three times: 1, 2, and 24 h after the addition of sodium fluorescein. The decrease in the optical density of the solutions from the apical side and the increase from the basolateral side of the cells were analyzed. The excitation and absorption wavelengths were 485 and 535 nm, respectively. The diffusion rates determined at 1, 2, and 24 h after the addition of fluorescein for each study group were pooled and compared.

#### 4.3.3. Western Blotting of the Cell Culture

IPEC-J2 monolayers were washed in PBS with Ca^2+^ and Mg^2+^ and lysed in RIPA buffer pH 7.6 (25 μM HEPES; 2 μM EDTA; 25 μM NaF; 1% SDS (10%)) and protease inhibitor (complete mini EDTA-free, Boehringer, Mannheim, Germany). Cells were scraped off the supports, pipetted into Eppendorfs, and placed in an ultrasonic bath for 8 seconds. Each sample was homogenized twice. Quantification, electrophoresis, and next steps of Western blot analysis were carried out as described earlier [56]. For densitometry, the detected protein bands were normalized using the Image Lab 6.1 Software (Bio-Rad, Hercules, CA, USA) on total protein amount.

### 4.4. Statistical Analysis

Statistical analysis was performed using the Anova group analysis in GraphPad Prism 6.01 (Graphpad Software Inc., San Diego, CA, USA). The data were analyzed using Mann–Whitney U-test (Figure 1 and Figure 4), and two-way ANOVA (Figure 3A), paired *t*-test (Figure 3B). The results of the analyses are presented as mean ± standard error (M ± SEM). Statistically reliable differences were reported with a probability value of 95% (*p* < 0.05).

## Figures and Tables

**Figure 1 ijms-22-10278-f001:**
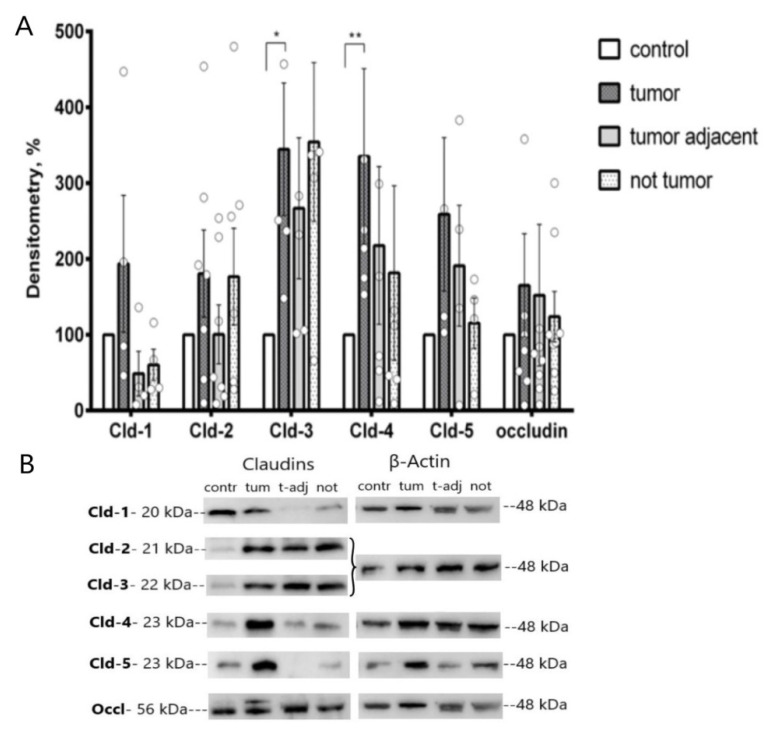
Western blot analysis of TJ protein expression in rat colon after chronic DMH administration. (**A**) densitometric analysis revealed increased claudin (Cld)-3 and -4 in tumor vs. control (*n* = 4 and *n* = 5, respectively). The values were normalized to β-actin as protein loading control. The number of symbols corresponds to the number of samples. (**B**) representative immunoblots, including stripped blots (the data for Cld-2 and Cld-3 had the same control. Cld-3 was determined after stripping of Cld-2, then β-actin was analyzed). Data (mean ± SEM); * *p* < 0.05, ** *p* < 0.01, Mann–Whitney U-test.

**Figure 2 ijms-22-10278-f002:**
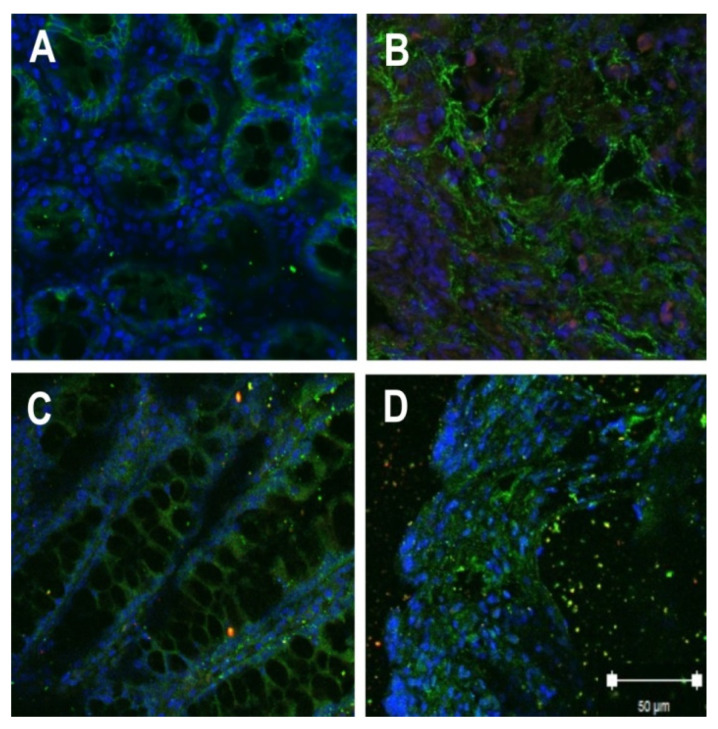
Localization of claudin-3 (**A**—control, **B**—tumor) and claudin-4 (**C**—control, **D**—tumor) in rat colon by immunofluorescence confocal microscopy (green); bar: 50 μm.

**Figure 3 ijms-22-10278-f003:**
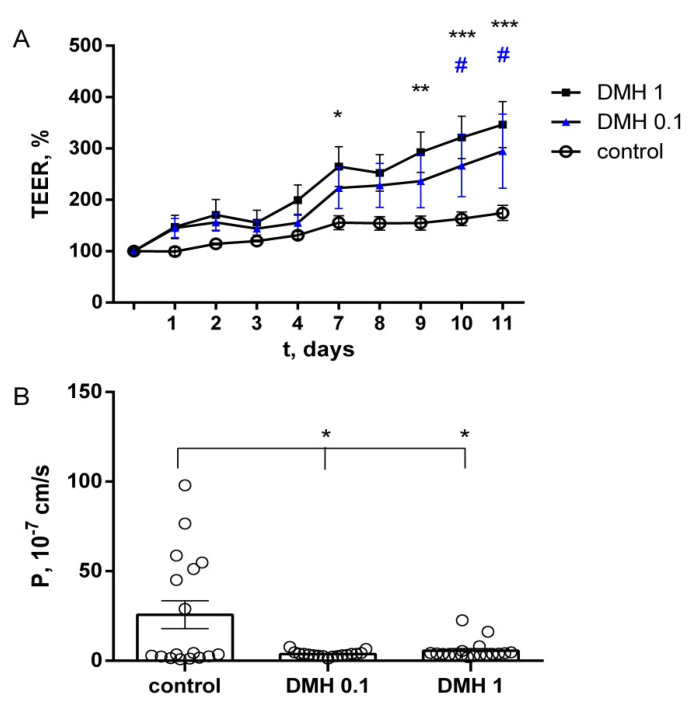
Cell monolayers. (**A**) transepithelial electrical resistance (TEER). * *p* DMH 1 < 0.05, ** *p* DMH 1 < 0.01, *** *p* DMH 1 < 0.001, # *p* DMH 0.1 < 0.05 compared with the control group, two-way ANOVA (n each group = 6). (**B**) permeability for fluorescein of the IPEC-J2 cell monolayers. * *p* < 0.05, paired *t*-test (n measurements for each group = 18).

**Figure 4 ijms-22-10278-f004:**
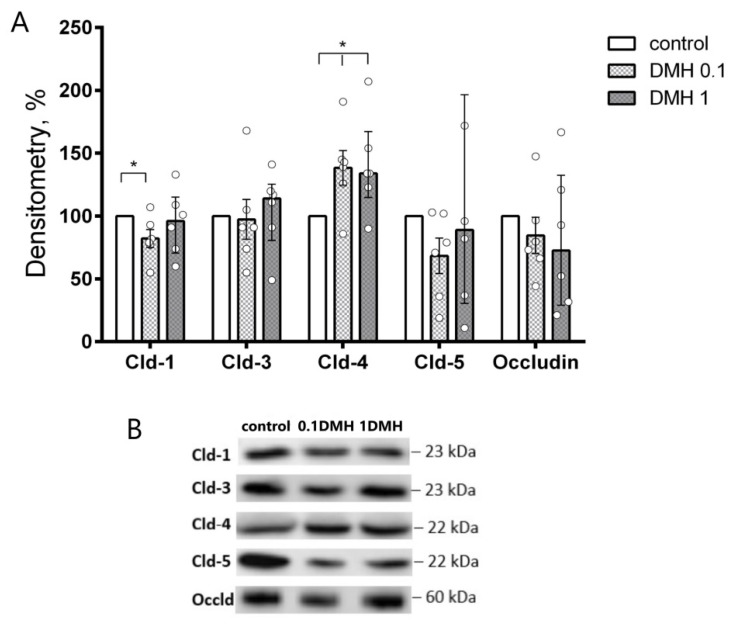
Western blot analysis of claudin expression in IPEC-J2 cell monolayers after 11 days incubation with DMH. (**A**) densitometric analysis revealed increased Cld-4 signals in both 0.1 and 1 μM DMH approaches vs. control, and decreased Cld-1 signals after incubation with 0.1 μM DMH (*n* = 6 in each group). The number of symbols corresponds to the number of samples. (**B**) representative Western blot bands. Data (mean ± SEM); * *p* < 0.05, Mann–Whitney U-test.

**Figure 5 ijms-22-10278-f005:**
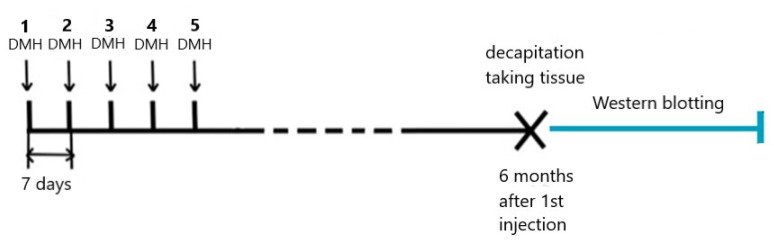
Animal experimental design (scheme).

## Data Availability

The data are contained in the article. The datasets analyzed in the study are available from the corresponding author upon reasonable request.

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
