# Peer review of "Effects of 1,2-Dimethylhydrazine on Barrier Properties of Rat Large Intestine and IPEC-J2 Cells"

_ijms, 2021, doi:10.3390/ijms221910278_

Round 1

Reviewer 1 Report

Effects of 1,2-dimethylhydrazine on barrier properties of rat large intestine and IPEC-J2 cells

Colon cancer is accompanied by a decrease of epithelial barrier properties which are determined by tight junction proteins between adjacent epithelial cells. The authors analyzed the expression of tight junction proteins in colon of rats with 1,2-dimethylhydrazine(DMH)-induced carcinogenesis, as well as the barrier properties and tight junction proteins expression of IPEC-J2 cells after incubation with DMH. Transepithelial electrical resistance and paracellular permeability for sodium fluorescein of IPEC-J2 monolayers was examined by epithelial volt/ohm meter and spectrophotometry. Further they analyzed the expression and showed the localization of TJ proteins by Western blot and immunohistochemistry. In the colon of rats with DMH-induced carcinogenesis, the level of claudin-3 and -4 in tumors was significantly increased compared to the control. Transepithelial electrical resistance of IPEC-J2 cells increased, while paracellular permeability for sodium fluorescein decreased, that was accompanied by an increased expression of claudin-4. The increase of claudin-4 in rat colon after chronic DMH exposure was consistent with the acute effect of DMH on IPEC-J2 cells that may indicate an essential role of this protein in colorectal cancer development. However, there are many critiques which needs to be addressed

  1. Please make a flow chart or schematic showing the animal study plan. Also include the details about how the animals were sacrificed.
  2. Please clear that the western blot in colon tissues were performed in the total extract or membrane fractions. It has been mentioned as membrane fractions, but the protocol looks like for total protein.
  3. In methods, it has been mentioned as immunohistochemistry but figure 2 looks like immunofluorescence staining. Please explain.
  4. In figure 4, loading control was missing. Please include that and reanalyze the densitometry with proper loading control.
  5. In figure 1, Beta actin for Cld-2 and Cld-3 looks same. Please check the same.
  6. DMH concentration in figures were mentioned as 0,1 instead of 0.1. Please change that everywhere.

Author Response

Dear Sir/Madam,

We thank you for your insightful and helpful comments. We have addressed each of these with revisions to the text of the manuscript, accordingly. If editing of English language and style would still be required, we will edit the manuscript using MDPI Author Services.

Point 1: Please make a flow chart or schematic showing the animal study plan. Also include the details about how the animals were sacrificed.

Response 1: We added Figure 5. Animal experimental design (scheme) to the Materials and Methods section and added: “Decapitation was rapidly accomplished in accordance with the recommendations of the AVMA [52]”. As it is a routine procedure, it may not require additional details. The main requirement is the very high speed of this manipulation and specially trained staff. It was necessary in our study design and it was approved by the ethics committee of St. Petersburg University, accordingly.

Point 2: Please clear that the western blot in colon tissues were performed in the total extract or membrane fractions. It has been mentioned as membrane fractions, but the protocol looks like for total protein.

Response 2: In the study, for lysis of colon tissues and IPEC-J2 cells we used the RIPA buffer, which ensures the isolation of proteins from the membrane fraction. We thank the reviewer for this important remark and added information regarding the RIPA buffer to the Materials and Methods section, accordingly.

Point 3: In methods, it has been mentioned as immunohistochemistry but figure 2 looks like immunofluorescence staining. Please explain.

Response 3: The fluorescence analysis based on immunofluorescence staining is a necessary part and the final stage of immunohistochemical study. We corrected the text by changing the title “Immunofluorescence staining“ to “Immunohistochemistry” in the methods section, and by correcting "fluorescence method" to "fluorescence analysis" in the results part.

Point 4: In figure 4, loading control was missing. Please include that and reanalyze the densitometry with proper loading control.

Response 4: The Stain-Free FastCast gels were used for the analysis of TJ proteins in the IPEC-J2 cells. Using the Stain-Free gels for immunoblotting, each protein is normalized on total protein amount, which is the loading control. Therefore, the use of a loading control like b-actin would not be necessary. We attached a file of the gel images with added images of total protein distribution.

Point 5: In figure 1, Beta actin for Cld-2 and Cld-3 looks same. Please check the same.

Response 5: Figure 4 includes the results for Cld-2 (anti-mouse antibody) and Cld-3 (anti-rabbit antibody) obtained from the same membrane; first, Cld-2 was determined, and then, after stripping, Cld-3. After the detection of the claudins, Beta actin was analyzed. Therefore, in this case, the control is the same. We have added this technical detail to the Figure legend, accordingly.

Point 6: DMH concentration in figures were mentioned as 0,1 instead of 0.1. Please change that everywhere.

Response 6: We have corrected this, accordingly.

Reviewer 2 Report

The manuscript is interesting, however revisions need to be carried out.

The authors need to go deeper into the introduction of MS, giving more examples, the state of the art of an MS is very important.

The MS discussion also needs to go deeper the topic of the article is important and the results need to be well explored.

Author Response

Dear Sir/Madam,

Thank you for your helpful and positive evaluation.

Point 1: The authors need to go deeper into the introduction of MS, giving more examples, the state of the art of an MS is very important. The MS discussion also needs to go deeper the topic of the article is important and the results need to be well explored.

Response 1: We have improved all parts of the revised manuscript, accordingly.

Round 2

Reviewer 1 Report

The authors has addressed all the comments and the manuscript looks good. However there is one concern that needs to be addressed before the acceptance of the manuscript.

Even though in figure 1  the blots were stripped and reused for Cld-3, which was fine and acceptable, just put the beta actin blot at one place only. Or rearrange the figures in such a way to show beta actin once. 

Author Response

Response to Reviewer 1 Comments

Dear Sir/Madam,

Thank you for your helpful and positive evaluation.

Point 1: Even though in figure 1 the blots were stripped and reused for Cld-3, which was fine and acceptable, just put the beta actin blot at one place only. Or rearrange the figures in such a way to show beta actin once.

Response 1: We have corrected this, accordingly: rearranged the figure and added some words to the figure caption.

Reviewer 2 Report

The authors performed a large review in MS, I recommend to publish

Author Response

Dear Sir/Madam,

thank you for your positive evaluation.

Kind regards,

Authors